# Study of Semi-Dry High Target Solidification/Stabilization of Harmful Impurities in Phosphogypsum by Modification

**DOI:** 10.3390/molecules27020462

**Published:** 2022-01-11

**Authors:** Fenghui Wu, Can Yang, Guangfei Qu, Liangliang Liu, Bangjin Chen, Shan Liu, Junyan Li, Yuanchuan Ren, Yuyi Yang

**Affiliations:** 1Faculty of Environmental Science and Engineering, Kunming University of Science and Technology, Kunming 650500, China; wufh2020@stu.kust.edu.cn (F.W.); cy2019@stu.kust.edu.cn (C.Y.); Liu2020@stu.kust.edu.cn (L.L.); Cbj2020@stu.kust.edu.cn (B.C.); LS2019@stu.kust.edu.cn (S.L.); LJYlab1978@stu.kust.edu.cn (J.L.); 2020REN@stu.kust.edu.cn (Y.R.); YYY2021@stu.kust.edu.cn (Y.Y.); 2National Regional Engineering Research Center-NCW, Kunming 650500, China

**Keywords:** phosphogypsum, modification, leaching toxicity, semi-dry, solidification/stabilization

## Abstract

Phosphogypsum (PG) treatment is one of the research hotspots in the field of environmental protection. Many researchers both at home and abroad have devoted themselves to studies on harmless resource treatment of PG, but the treatment technology is unable to meet the demand of PG consumption due to the huge production and storage demands. In order to solve the problem of PG pollution, this study explored the different solidified effects of various modification formulations on the hazardous components in PG, using industrial solid waste calcium carbide slag (CCS) as an alkaline regulator; Portland cement (PC), polyaluminum chloride (PAC) and CaCl_2_ as the main raw materials of the solidification and stabilization formula and the water content in PG as the reaction medium. The results showed that CCS (0.5%), PC (0.4%) and PAC (0.3%) had a more significant solidified effect on phosphorus (P) and fluoride (F). PAC was added in two steps and reacted under normal temperature and pressure, and its leaching toxicity meets the requirements of relevant standards, which laid an excellent foundation for PG-based ecological restoration materials and filling materials, with low economic cost, simple process and strong feasibility. This will provide great convenience for the later mining and metallurgy.

## 1. Introduction

With increased attention being paid to the world’s ecological environment, various policies and regulations on environmental protection have been issued [1,2]. Phosphogypsum (PG) is one of the most studied solid wastes in the field of environmental protection [3]. Globally, around 170 million tons of PG are disposed of every year. The total production of PG is expected to reach about 7–8 billion tons by 2025 (International Atomic Energy Agency). With the growing world population, the demand for phosphoric acid in agriculture [4], chemical industry, national defense and other fields is increasing [5]. Mineral resources are gradually depleted, showing a gradual decrease in high-grade ores [6], but the yield of PG and the purity of phosphoric acid mainly depend on the grade of phosphate rock [6]. The production of 1 ton phosphoric acid can produce 4–6 tons of PG. With the gradual decrease in phosphate rock grade, the unit production of PG will gradually increase [6]. The global annual emission of PG is about 120 million tons, of which 70 million tons per year are produced in China, with reserves of more than 250 million tons. In addition, there will be increasingly relevant requirement levels of PG disposal, in most cases without pretreatment, which can cause serious environmental contamination due to impurities. This contamination may result from the heavy metal, phosphorus (P) and fluoride (F) [7,8] attached to the surface of the calcium sulfate dihydrate crystal, fluorine-containing phosphorus crystal water [9,10,11], capillary water and attached water. Therefore, solving the environmental pollution problem of PG is an urgent task [3,12,13].

In recent years, many environmentalists have attached great importance to the resource treatment of PG and have been eager to solve this problem. The current studies on resource utilization of PG are primarily focused on the following aspects: preparation of sulfuric acid and cement [14], construction materials [15], improvement of soil [16], and synthesis of high-strength gypsum [17], CaCO_4_ [18], calcium sulfate whiskers [19], agricultural fertilizers [20,21,22], soil conditioners and so on [23,24,25,26]. Especially, much of the work revealed the great opportunities related to the use of PG in different applications. However, in most of the relevant studies mentioned above, the consumption of PG was too small to meet the purpose of large-scale consumption. Some of these studies meet the demand, but some pretreatment of the PG must be performed, such as thermal treatment or washing, which can consume a large amount of energy and resources. Secondary pollution and increased costs can easily occur [27]. To date, many studies have been carried out on the resource utilization of PG, but the production and inventory of PG are still huge. Due to the cost, technical maturity and other reasons, the resource utilization of PG cannot be popularized and applied on a large scale [28]. 

To sum up, although there are many experts and scholars working on PG treatment, they are still unable to utilize a large number of disposal technologies. At present, the barren soil leads to the reduction in grain production, and a large number of mines and caves left by mining activities are prone to geological disasters. PG as soil or mine filling material has broad application prospects for non-hazardous treatment. There have been many studies on mine slurry filling, including Rong’s hemihydrate gypsum filling [29], and Zhang’s desulfurized gypsum filling [30]. It can be seen that filling technology is one of the effective methods to quickly absorb a large amount of PG. The research aims to reduce the cost of PG treatment and reduce environmental pollution risk. To achieve this aim, a semi-dry method to solidify and stabilize PG was used. Considering the types of curing agent, curing method, curing time and cost of raw materials, a new PG treatment technology for mine filling is proposed. It provides data and technical support for the large-scale consumption of PG. It is also a technical means that can be applied to practical engineering, reduces the environmental risks from PG and mitigates natural disasters left behind by mining.

## 2. Raw Materials and Experimental Methods

### 2.1. Raw Materials

Phosphogypsum (PG) was obtained from Yunnan Yuntianhua Environmental Protection Technology Co., Ltd., Wuhan, China. Calcium carbide slag (CCS) was obtained from Yunnan Energy Investment Group Co., Ltd., Kunming, China. Polyaluminum chloride (PAC) (analytical purity, AR) was purchased from Jinyuxing Chemical Co., Ltd, Beijing, China. Calcium chloride (CaCl_2_) (analytical purity, AR) was purchased from Wanruida Chemical Co., Ltd., Hong Kong, China. Portland cement (PC) was purchased from Jinhua Chenming Building Materials Co., Ltd, Jinhua, China, and a pure water machine (ULPHW-IV) was purchased from Sichuan YOUPU Ultrapure Technology Co., Ltd., Chengdu, China. All companies listed are located in China.

### 2.2. Sample Preparation

In order to explore the solidification of harmful substances in PG, the main steps were as follows: (1) Using the alkalinity of CCS to neutralize the acidity of PG, the effects of different addition amounts of CCS on pH, heavy metals, P and F leaching concentration were explored to find the optimal ratio. The details of proportions are shown in Table 1. The curing times are 1, 3 and 5 days, and the influences of CCS on impurities in PG were tested.

(2) On the basis of (1), the optimum dosage of CCS was explored. The solidification and stabilization effects of harmful substances were explored by adding small amounts of PC, CaCl_2_ and PAC as solidified agents for hazardous substances. The detailed ratios are shown in Table 2, and the curing times are 1, 3 and 5 days.

### 2.3. Testing Methods

#### 2.3.1. XRF

The PG was dried in a drying oven in air at 105 °C for 24 h. The elemental chemical composition was then obtained by X-ray fluorescence spectrometry (ARL PERFORM’X), The equipment was purchased from Thermo Fisher Scientific, Waltham, MA, USA.

#### 2.3.2. XRD

The phases of samples were examined using an X-ray powder diffractometer (Physics Ultima IV), which was purchased from Bright Industrial Co., Ltd. (Shanghai, China), with Cu -Kα radiation (λ = 0.154060 nm) at a scanning rate of 1.56° per min in the 2θ range from 4° to 80°.

#### 2.3.3. SEM-EDS

The crystal morphology was observed using a field emission scanning electron microscope (ZEISS Ce mini 300, OXFORD Xplore), which was purchased from Zeiss optical instruments (Shanghai, China) International Trade Co., Ltd. The elemental measurements of the products were determined by an energy-dispersive spectrometry device.

#### 2.3.4. FTIR

Variations of the active groups in the sludge filtrate and filter cake were analyzed using a Fourier transform infrared analyzer (Thermo Scientific Nicolet iS50), which was purchased from Thermo Fisher Scientific, USA. Listed below are the analytical conditions: signal-to-noise, 500:1; scan speed, 40 sheets/s; resolution, 2 cm^−1^; wavenumber range, 500 to 4000 cm^−^^1^; integral number, 10 times; measurement time, 10 s.

#### 2.3.5. Particle Size Analysis

A laser particle size analyzer (LT3600 plus) purchased from Zhuhai truth Optical Instrument Co., Ltd., Zhuhai, China, was used for particle size analysis.

#### 2.3.6. Thermogravimetric Analysis

A thermogravimetric analyzer was purchased from METTLER TOLEDO International Trading (Shanghai, China) Co., Ltd. The TG/DTA (TGA 209 F1) curves were recorded in the range of 20 to 800 °C with a heating speed of 10 K·min^−1^.

#### 2.3.7. Leaching Toxicity Test

According to the standard of the horizontal oscillation method for leaching toxicity of solid waste (HJ 557-2010), the leaching toxicity experiments of PG and solidified PG were carried out. The P and F were tested by ion chromatography (ion chromatography device, Model IC1800, Sunny Hengping Co., Ltd., Shanghai, China), the heavy metals were tested using an atomic absorption spectrometry (atomic absorption spectrophotometer, model AA-7020, purchased from Beijing East West Analytical Instrument Co., Ltd., Beijing, China) and the acidity and alkalinity were tested with a pH tester (Leici phs-3c desktop), purchased from Jinan Oulaibo Scientific Instrument Co., Ltd., Jinan, China. The indexes were compared with the maximum concentration limits of the comprehensive discharge standard for sewage (GB8978-1996).

#### 2.3.8. Radioactivity Test

A low background α/β radioactivity meter was used for the α/β radioactivity test. The measuring instrument (PAB 3000 Ⅱ) was purchased from Wuhan Puxie Technology Co., Ltd., Wuhan, China.

## 3. Results and Discussion

### 3.1. The Physicochemical Properties of CCS and PG

The moisture content of PG is 28.98%, pH = 2.77, the water-soluble salt content is 0.877% and the organic matter content is 0.225%. The moisture of CCS is 26.76%, pH = 11.79, the water-soluble salt content is 1.808% and the organic matter content is 1.05%. It is feasible to modify PG by using CCS in terms of the acidity of the raw materials.

As shown in Table 3 from XRF analysis, it can be concluded that the main oxide components of PG and CCS are CaO. CaO content and SO_3_ content in PG show little difference, which may be due to the presence of CaSO_4_ or CaSO_4_·nH_2_O. The content of CaO in CCS is about 88%, while the content of other substances is smaller, indicating that the main components of CCS may be CaO, Ca(OH)_2_ or CaO·nH_2_O. The exact composition needs to be further determined.

As shown in Figure 1, The SEM of PG and CCS shows that PG mainly forms a parallelogram sheet stacking structure, which contains many impurities, and most of them are fractured. The EDS analysis shows that the main elements on the surface are Ca, O, S and some rare elements. The micromorphology of CCS is mainly amorphous with irregular agglomeration, irregular columnar and acicular. EDS characterization indicates that the major surface elements are Ca, O, Na, Al, Si, S and Cl.

As shown in Figure 2, the main component of PG is CaSO_4_·2H_2_O (78–90%), and the main component of CCS is Ca(OH)_2_ (80–95%).

As shown in Figure 3, the particle size distribution ranges of PG and CCS are similar (0.5–800 μm), but the particle size distribution of PG has a certain regularity and is relatively uniform, while the particle size distribution of CCS is not very uniform.

The TG-DSC analysis (Figure 4) shows that there are two endothermic peaks in both PG and CCS. Since the moisture content of both PG and CCS is more than 20%, and there is crystal water in PG, CCS is dehydrated by Ca(OH)_2_, and water and volatile substances are firstly volatilized from PG and CCS in the process of gradual heating. The evaporation temperature of crystal water in CCS is higher than that of PG. With the increase in temperature, the reaction of PG and CCS is shown in the Formulas (1) and (2).
(1)$${\mathrm{CaSO}}_{4}\cdot 2H_{2}O\overset{▲}{\to }{\mathrm{CaSO}}_{4}\cdot 0.5H_{2}O+1.5H_{2}O\overset{▲}{\to }{\mathrm{CaSO}}_{4}+2H_{2}O$$
(2)$$\mathrm{Ca}(\mathrm{OH}{)}_{2}\overset{▲}{\to }\mathrm{CaO}+H_{2}O$$

Based on the above analysis of the physicochemical properties of raw materials, it is concluded that it is feasible for CCS to neutralize the acidity of PG, and it has a certain moisture content. Without adding water, PG reacts with CCS to adjust the pH to achieve waste treatment. 

### 3.2. The Effect of Different CCS Addition on Leaching Toxicity of PG

We added different dosages of CCS to neutralize the PG and leached according to the standard of the “horizontal oscillation method for leaching toxicity of solid waste” (HJ 557-2010), and the pH value of the leaching solution was measured.

As shown in Figure 5, the pH change of leaching solution was tested as the addition of CCS decreased from 20% to 0.1%. The pH value of the PG leaching solution first remained stable, then decreased slowly and finally decreased sharply with the decrease in CCS addition. When adding CCS of more than 5%, the range of pH change was not very large, and the pH value stayed above 11. With the gradual reduction in CCS addition, the pH value when adding 0.70–2.5% CCS was reduced to 10–11. With the further reduction in CCS addition, the pH values of CCS added at 0.6% and 0.5%, respectively, were 6–9, which were in accordance with the relevant standards. When the addition of CCS was less than 0.40%, the pH was below 6.

The influence of different additions of CCS on the leaching toxicity of PG refers to the leaching concentration of water-soluble salt, and the results are shown in Figure 6.

In general, the water-soluble salt does not change much after curing for 1, 3 and 5 days. Specifically, the water-soluble salts after curing for 1 day are only slightly higher than that after curing for 3 days, and those after 3 days of curing are higher than after curing for 5 days. When the CCS content is 0.9% and 0.8%, the water-soluble salt content is the lowest after 1 day of curing, and the pH value is in the range of 11 to 12. The content of water-soluble salt is less than 2% in the range of pH 6 to 9.

As shown in Figure 7, the overall trend of heavy metal content in the leaching solution increases gradually with the decrease in CCS content, and the metal Mn content increases sharply at pH 11–12. In contrast, the leaching concentrations of heavy metals are relatively low and meet the requirements of relevant concentration limits in “integrated wastewater discharge standard” (GB8978-1996), while it can be seen that the leaching efficiency of most heavy metals increases as the pH decreases. The PO_4_^3−^ and F^−^ increased with the decrease in pH, and the leaching concentration of PO_4_^3−^ was higher than the limit value of 0.5 mg/L in the comprehensive discharge standard for sewage (GB8978-1996). The concentration of F is less than 10 mg/L when the CCS of more than 5% is added, but it is close to 10 mg/L when the pH is 6–9.

The sample containing 1% CCS was further characterized to explore the effect mechanism of CCS and PG. As shown in Figure 8, the addition of 1% CCS leads to subtle changes in PG, and the micro morphology of PG has gradually changed from parallelogram block to hexagonal prism. At the same time, the phenomenon of adsorption agglomeration of PG crystals and the clumping of multiple PG crystals was found. It can be inferred that the addition of CCS caused a local reaction. From the surface element analysis, the major element composition of PG did not change, which may be due to the fact that the EDS characterizes the surface element and the major elemental composition of the selected surface of PG after the agglomeration is still Ca, S, O, C.

As seen in the XRD characterization, the main component of PG is still CaSO_4_·2H_2_O. Compared with PG, calcium silicate, orthophosphate, calcium fluoride and other new components appear when 1% CCS is added. The FTIR characterization further demonstrates the significant solidification of P and F in PG with a strong absorption peak at 1110 cm^−1^ and a weak absorption peak at 600 cm^−1^, showing the symmetry and stretching peak of the phosphate ion. There are O-H vibration peaks in the wavelength range of 3400 to 3500 cm^−1^, which is caused by the O-H vibration in 3Ca_2_(PO4)_2_ and Ca(OH)_2_, a weak F-Ca peak at 1140 cm^−1^ indicate the formation of CaF_2_.

From the above characterization, it can be inferred that the addition of carbide slag may cause the following reactions (3)–(11).
H_2_SiO_3_ + CaO = CaSiO_3_ + H_2_O(3)
SiO_2_ + Ca(OH)_2_ = CaSiO_3_(4)
MO + Si + 2CaO = 2M + Ca_2_SiO_4_(5)
CaMg(CO_3_)_2_ + H^+^ + 2F^−^ → (Ca, Mg, Zn)F_2_ + CO_2_ + H_2_O(6)
CaSO_4_·2H_2_O + 2F^−^ → CaF_2_ + SO_4_^2−^ + 2H_2_O(7)
(Ca^2+^, Mg^2+^, Zn^2+^) + 2OH^−^ → (Ca, Mg, Zn)(OH)_2_(8)
(Ca^2+^, Mg^2+^, Zn^2+^) + 2F^−^ → (Ca, Mg, Zn)F_2_(9)
(Ca, Mg, Zn)(OH)_2_ + PO_4_^3−^ → (Ca, Mg, Zn)PO_3_(OH)·2H_2_O(10)
(Ca^2+^, Mg^2+^, Zn^2+^) + PO_4_^3−^ → (Ca, Mg, Zn)_3_(PO4)_2_(11)

### 3.3. The Solidification Effect of Different Formulations on Harmful Components in PG

#### 3.3.1. pH

PG was used as the main raw material, CCS was added to adjust its pH value, PC and PAC were used as the curing agents of harmful substances and CaCl_2_ was used as the fluorine curing intensifier. PG was modified according to the formula and proportions shown in Table 2 and maintained for 1, 3 and 5 days. The leaching toxicity test was carried out according to the standard of the horizontal oscillation method for leaching toxicity of solid waste (HJ 557-2010). Firstly, the pH changes of different formulations and curing times were tested.

As shown in Figure 9, the pH test results were compared with the pH standard value in “the comprehensive discharge standard for sewage” (GB8978-1996). It was found that the pH meets the requirements of the standard discharge limits when adding 0.5% CCS, 0.04% PC, 0.15% PAC or without adding PC, adding 0.5% CCS, 0.1% CaCl_2_, 0.15% PAC and 0.3% PAC. At the same time, we found that the pH did not change much whether the curing time was 1, 3 or 5 days, indicating that the reaction time between the powders was within 1d.

#### 3.3.2. Heavy Metal

Combined with the analysis of the modification of PG by the CCS, the curing time had little effect on the toxicity of leaching. The next modification was to select the modified materials for 1 day for analysis.

As shown in Figure 10, the leaching concentrations of Ni, Cd, Cu, Pb, Mn, Cr and Cr^6+^ in PG were analyzed and compared with the most stringent concentration limits in “the integrated wastewater discharge standard” (GB8978-1996). It was found that the leaching toxicity of heavy metals modified by PG met the requirements. Although the leaching concentrations of Cu, Pb and Mn were higher than those of other metals, they met the requirements of the standard. It can be seen that under normal conditions, the threat of heavy metals in PG is not very serious.

#### 3.3.3. P, F, Cl, Ammonia Nitrogen and Water-Soluble Salt

In order to further explore the potential pollution threat of modified PG, we conducted leaching concentration experiments of P, F, ammonia nitrogen and water-soluble salts to ensure the environmental safety performance of modified PG.

As shown in Figure 11, in terms of PO_4_^3−^ leaching concentration, except formula No. 22 and 23, the most stringent discharge concentrations specified in the comprehensive discharge standard for sewage (GB8978-1996) and class I solid waste requirements specified in pollution control standard for storage and landfill of general industrial solid waste (GB18599-2020) are met. For the leaching toxicity of F^−^, only formulas 24 and 28 meet the requirements of the first two standards. The results show that the leaching concentrations of water-soluble salts and ammonia nitrogen vary greatly, but they all meet the requirements of “class I solid waste” specified in “the pollution control standard for storage and landfill of general industrial solid waste” (GB18599-2020). To sum up, the overall indicators meet the most stringent discharge concentrations specified in “comprehensive discharge standard for sewage” (GB8978-1996) and “class I solid waste” requirements specified in “the pollution control standard for storage and landfill of general industrial solid waste” (GB18599-2020), formulations 24 and 28 in table III show significant stabilization effects, that is, PG content of 99.16%, CCS addition of 0.5%, PC of 0.04%, PAC of 0.3% and PG of 99.10%, CCS 0.5%, CaCl_2_ 0.1%, PAC 0.3%.

It can be seen from the SEM characterization Figure 12A–D that when the addition of carbide slag and PC is 0.5% and 0.1%, respectively, the addition of PAC changes from 0.06% to 0.30%, and the crystal morphology of PG changes obviously, gradually producing more agglomerated substances. The original crystal form of PG changes, indicating that the addition of CCS, PC and PAC lead to the reaction with PG. In the reaction process, the pH of the PG system was improved, and P, F, heavy metals and other substances were solidified. At the same time, The EDS characterization of formulas 24 and 28, which exhibited better curing results, was carried out, and the main surface elements were Ca, Al, Si, O, etc., indicating the formation of ettringite (3CaO·Al_2_O_3_·3CaSO_4_·32H_2_O) [31] during the reaction. Moreover, with the increase in Al content in PAC, the amount of ettringite also increases gradually. Its cementitious activity solidifies some toxic and harmful elements such as P, F and heavy metals and stimulates the cementitious activity of some PG. Combined with XRD analysis, it can be seen from A to D (Figure 12) that the diffraction peak of calcium sulfate dihydrate changed obviously. Through Jade 9 analysis, the new peaks mainly include CaF_2_, Ca_3_(PO_4_)_2_, ettringite, etc. [32], and with the increase in PAC addition, the more obvious the peak, the more new substances are generated [33]. In the SEM characterization of E–I (Figure 12), there are many kinds of morphologies, including hexagonal prism, combined with PG with only CCS. The hexagonal crystal is generated by CCS initiation, while the flocculent material generated by H is formed by PAC and water reaction. Combined with the XRD characterization of E–I (Figure 12), there is no obvious change in A–D (Figure 12), indicating that the flourishing chemical reaction is accompanied by physical changes. FTIR shows that the stretching vibration peaks of the O–H bond of calcium orthophosphate are in the range of 3400 to 3600 cm^−1^, with two obvious peaks [34]. The presence of two strong anti-symmetric stretching peaks for phosphate in the wave number range of 1050 to 1150 cm^−1^ and weak asymmetric corner peaks for phosphate in the range of 540 to 630 cm^−1^ further demonstrate the existence of orthophosphate. Because PAC was added to A–I (Figure 12), the same substances were produced, but the amount was changed. The two peaks at the wavenumber of 1000–1200cm^−^^1^ indicate the vibration peak of F-Ca, indicating the existence of CaF_2_ material. Meanwhile, ettringite may also be formed, but it cannot be identified because of the overlapping peaks [35].

#### 3.3.4. Radioactivity

Since some PG is radioactive in many regions, we studied the radioactivity of PG after stabilization and tested the radioactivity of the leached solution after leaching.

As shown in Figure 13, the leaching solution of modified PG, the total α-radioactivity was less than 1 Bq/L and the total β-radioactivity was less than 1 Bq/L. The radioactivity is less than 10 Bq/L, which meets the requirements of “the comprehensive discharge standard for sewage” (GB8978-1996).

## 4. Conclusions

The semi-dry method was used to stabilize and solidify PG. Firstly, industrial solid waste CCS was used to neutralize its acidity and alkalinity. When the dosage of CCS was 5%, the leaching toxicity of PG modified by CCS was analyzed according to the standard of the horizontal oscillation method for leaching toxicity of solid waste (HJ 557-2010). The pH of leaching solution was in the range of 6 to 9 and meets the requirements of “class I solid waste” specified in the pollution control standard for storage and landfill of general industrial solid waste (GB 18599-2020).

With contents of CCS 0.5%, CP 0.04%, first PAC 0.2%, second PAC 0.1%; or CCS 0.5%, CaCl_2_ 0.04%, first PAC 0.2%, second PAC 0.1%, the leaching toxicity indexes of the modified PG meet the relevant requirements of “class I solid waste” in the standard for pollution control of general industrial solid waste storage and landfill (GB 18599-2020).

The modified PG meets the requirements of environmental protection filling and ecological restoration materials and provides convenient conditions for secondary mining.

## Figures and Tables

**Figure 1 molecules-27-00462-f001:**
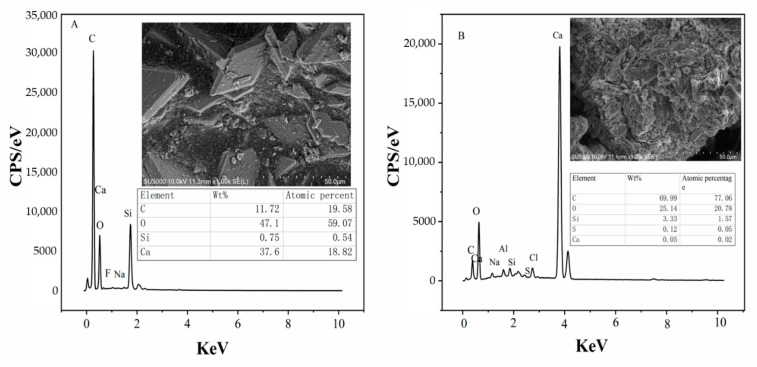
SEM-EDS of raw materials ((**A**)—PG, (**B**)—CCS).

**Figure 2 molecules-27-00462-f002:**
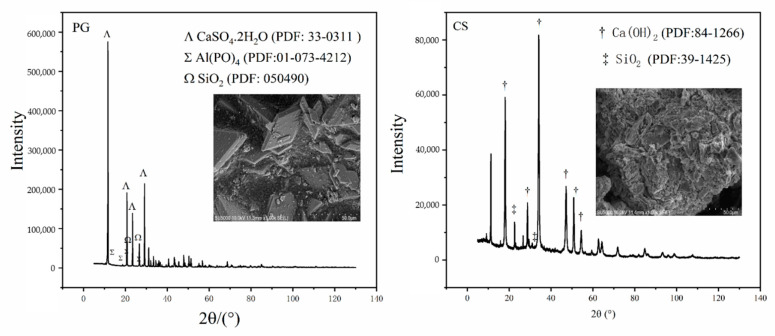
XRD of raw materials.

**Figure 3 molecules-27-00462-f003:**
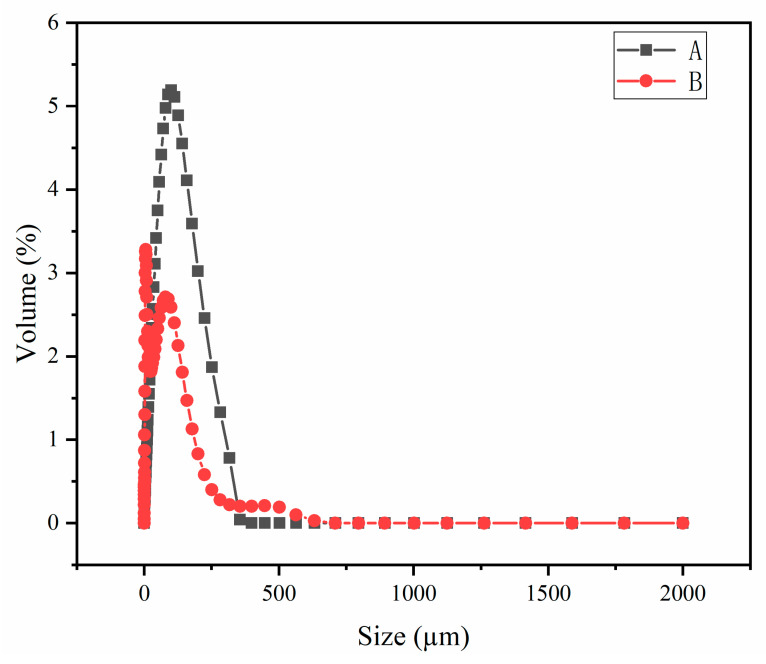
Particle size distribution of raw materials ((**A**)—PG, (**B**)—CCS).

**Figure 4 molecules-27-00462-f004:**
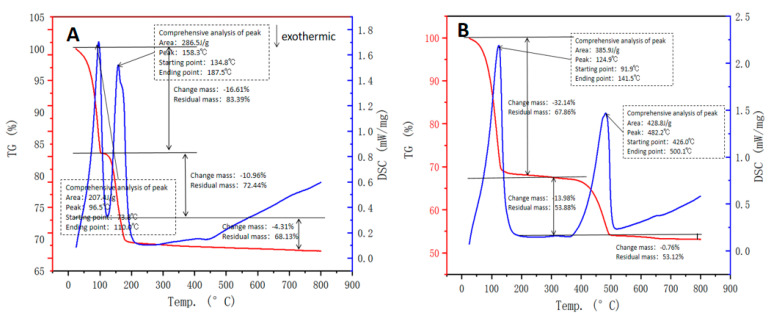
TG-DSC of raw materials ((**A**)—PG, (**B**)—CCS).

**Figure 5 molecules-27-00462-f005:**
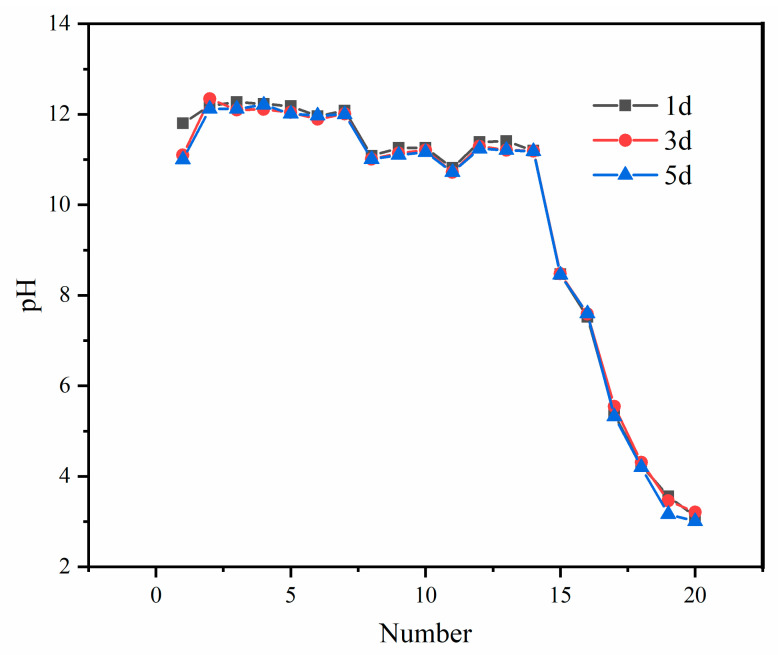
pH of PG after neutralization.

**Figure 6 molecules-27-00462-f006:**
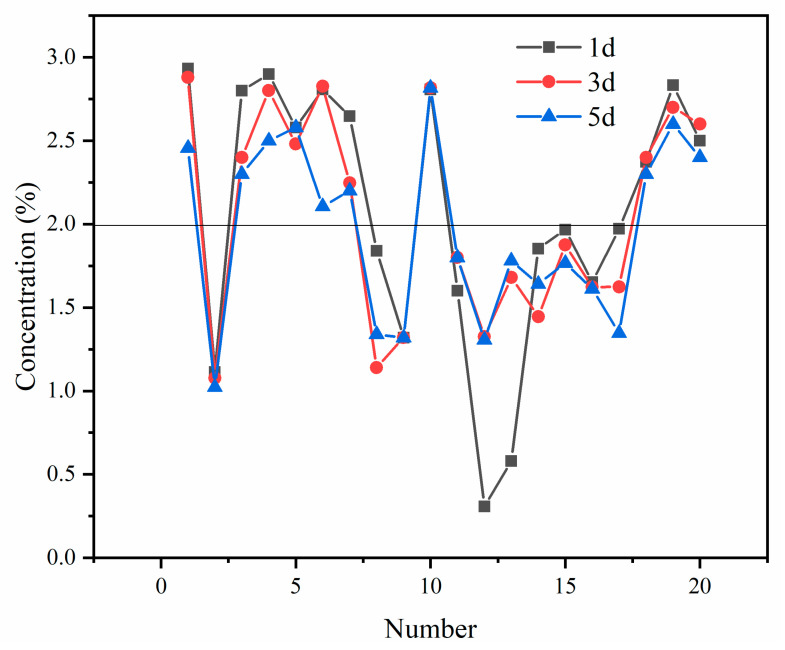
Variety of water-soluble salts.

**Figure 7 molecules-27-00462-f007:**
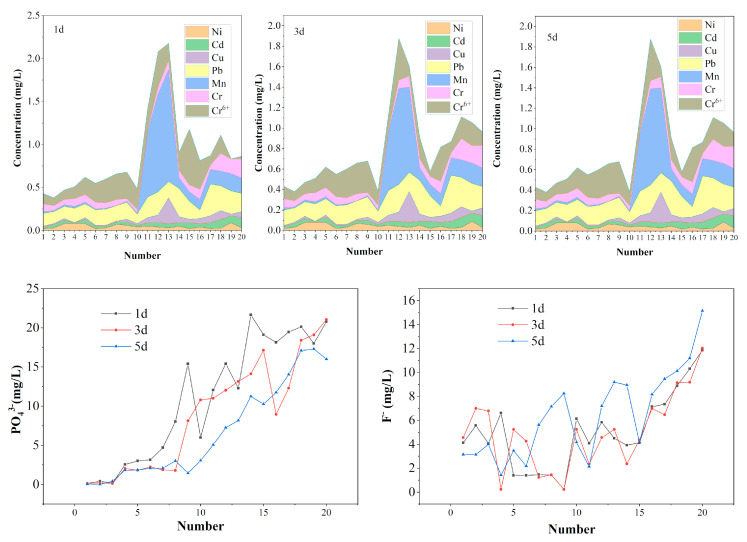
Leaching concentration of harmful substances.

**Figure 8 molecules-27-00462-f008:**
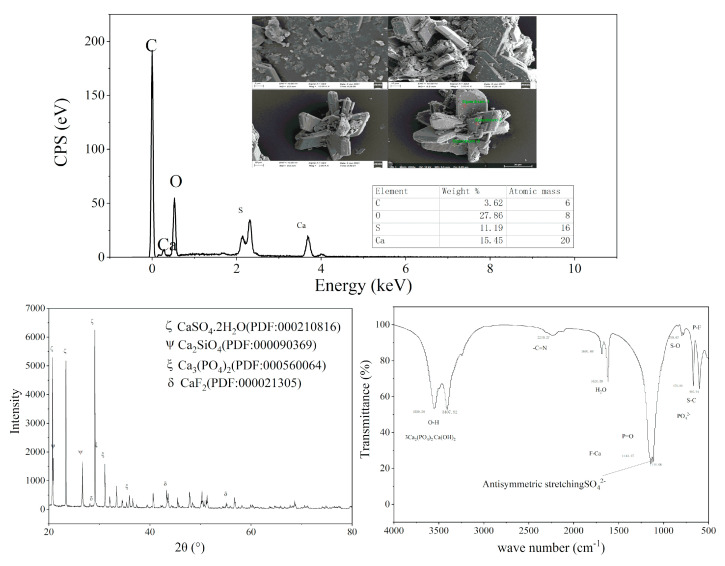
SEM-EDS, XRD and FTIR with addition of 1% CCS.

**Figure 9 molecules-27-00462-f009:**
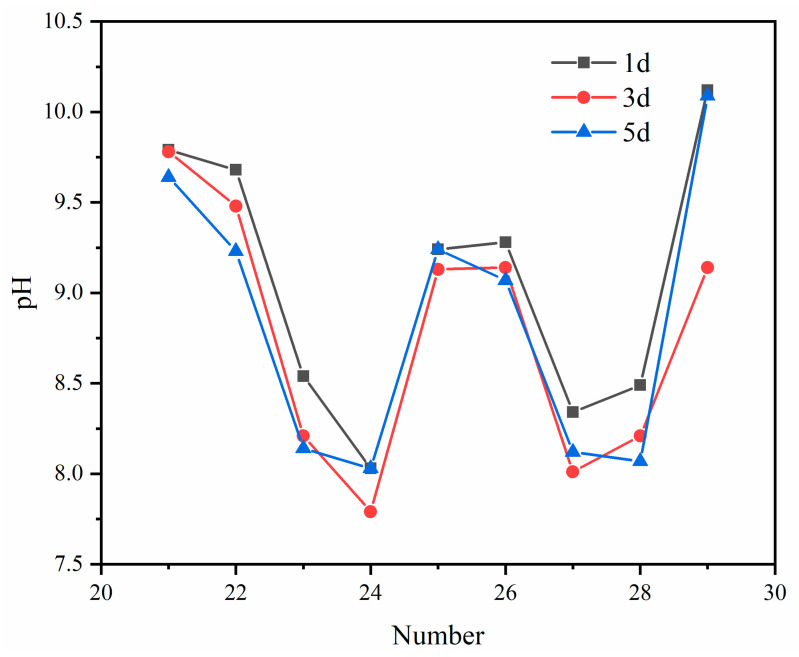
The pH change of different formulas and times.

**Figure 10 molecules-27-00462-f010:**
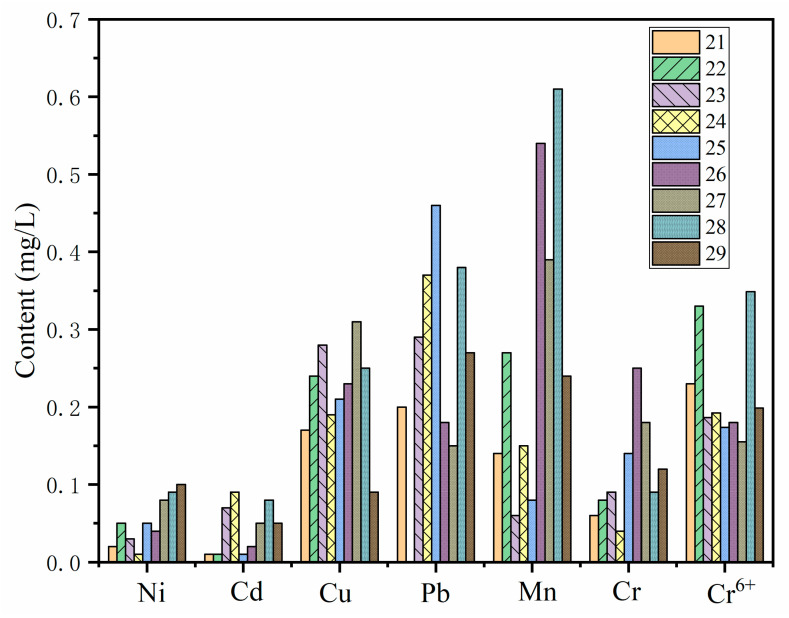
Leaching toxicity of heavy metals.

**Figure 11 molecules-27-00462-f011:**
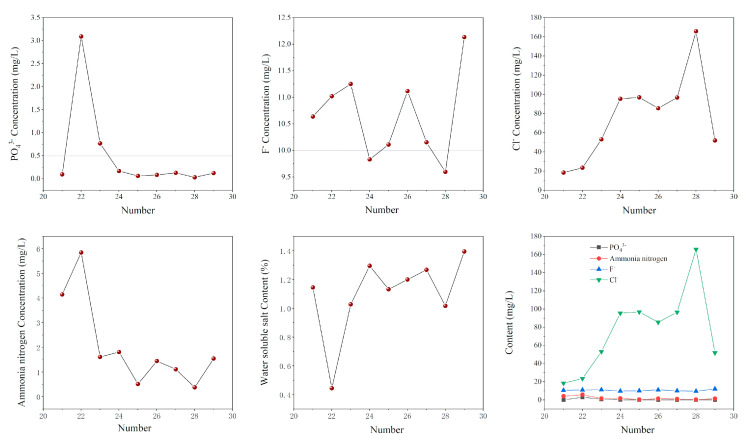
The concentration of P, F, Cl, ammonia nitrogen and water-soluble salt.

**Figure 12 molecules-27-00462-f012:**
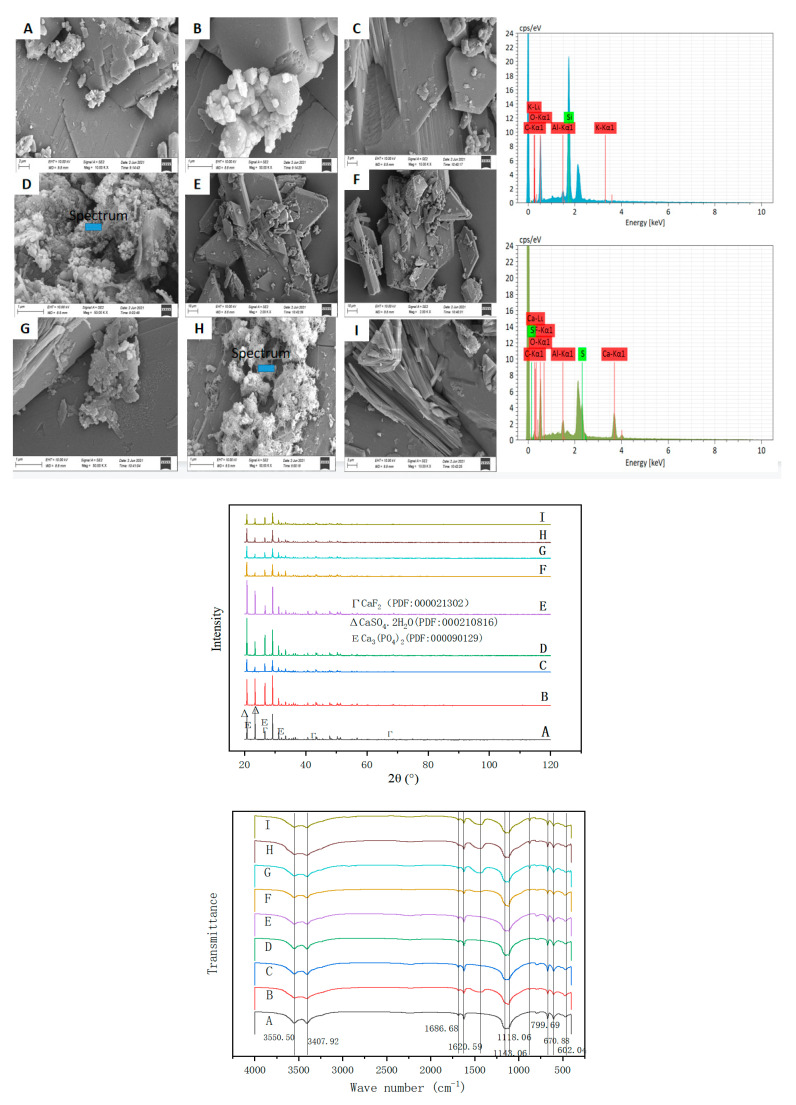
Characterization of different formulations by SEM-EDS, XRD and FTIR. ((**A**)—21, (**B**)—22, (**C**)—23, (**D**)—24, (**E**)—25, (**F**)—26, (**G**)—27, (**H**)—28, (**I**)—29).

**Figure 13 molecules-27-00462-f013:**
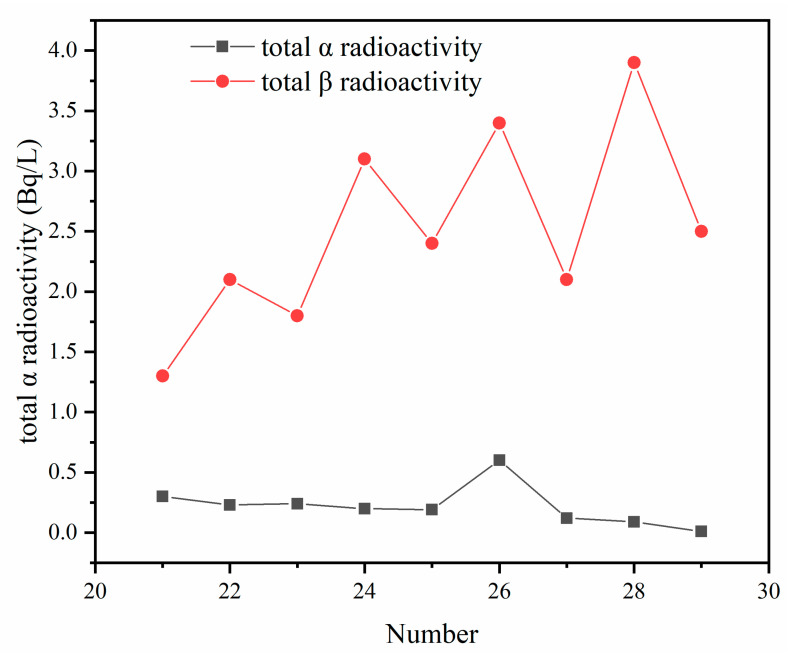
The total content of leaching solution with different formula α and β radioactivity.

**Table 1 molecules-27-00462-t001:** Different ratios of CCS and PG.

**No.**	**1**	**2**	**3**	**4**	**5**	**6**	**7**	**8**	**9**	**10**
PG	80.00%	82.50%	85.00%	87.50%	90.00%	92.50%	95.00%	97.50%	98.00%	98.50%
CCS	20.00%	17.50%	15.00%	12.50%	10.00%	7.50%	5.00%	2.50%	2.00%	1.50%
**No.**	**11**	**12**	**13**	**14**	**15**	**16**	**17**	**18**	**19**	**20**
PG	99.00%	99.10%	99.20%	99.30%	99.40%	99.50%	99.60%	99.70%	99.80%	99.90%
CCS	1.00%	0.90%	0.80%	0.70%	0.60%	0.50%	0.40%	0.30%	0.20%	0.10%

**Table 2 molecules-27-00462-t002:** Different ratios of CCS and PG.

No.	PG	CCS	PC	CaCl_2_	PAC
**21**	99.41%	0.50%	0.04%	/	0.06%
**22**	99.39%	0.50%	0.04%	/	0.08%
**23**	99.31%	0.50%	0.04%	/	0.15%
**24**	99.16%	0.50%	0.04%	/	0.30%
**25**	99.35%	0.50%	/	0.10%	0.06%
**26**	99.33%	0.50%	/	0.10%	0.08%
**27**	99.25%	0.50%	/	0.10%	0.15%
**28**	99.10%	0.50%	/	0.10%	0.30%
**29**	99.34%	0.50%	0.02%	0.04%	0.11%

**Table 3 molecules-27-00462-t003:** Analysis of XRD of PG and CCS.

**PG**	**CaO**	**SO_3_**	**SiO_2_**	**P_2_O_5_**	**Al_2_O_3_**	**F**	**MgO**	**K_2_O**	**Na_2_O**	**Fe_2_O_3_**	**Other**
Content (Wt%)	40.31	40.25	12.82	2.24	1.58	1.31	0.404	0.255	0.248	0.181	0.0934
**CCS**	**CaO**	**SiO_2_**	**Al_2_O_3_**	**Cl**	**Na_2_O**	**SO_3_**	**F**	**ZnO**	**Fe_2_O_3_**	**MgO**	**Other**
Content (Wt%)	88.32	3.57	2.72	2.04	1.30	0.726	0.39	0.244	0.237	0.235	0.0053

## Data Availability

Not applicable.

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
