# Peer review of "Study of Semi-Dry High Target Solidification/Stabilization of Harmful Impurities in Phosphogypsum by Modification"

_molecules, 2022, doi:10.3390/molecules27020462_

Round 1

Reviewer 1 Report

The topic of the phosphogypsum use of different genesis is very relevant today in different countries of the world. This study is devoted to an important environmental issue, because phosphogypsum contains a number of harmful impurities, which can limit its use and requires additional treatment.

At the end of the Introduction section, the purpose and objectives of the study should be clearly identified. Example: “The research aims to …. To achieve this aim, the following tasks were set: ...”

In the used methods of testing (item 2.3) when mentioning the equipment on which the analysis of samples was carried out, it is necessary to write in brackets the manufacturers of this equipment and the country of the manufacturer.

123, 177, 282, 288, 290, 292, 347, 351: The standards with which the author of the manuscript compared the results obtained are Chinese standards, is it possible to expand and specify also international standards for comparison?

In the Results and discussion section (3) it is necessary when discussing the results obtained to make reference to previous studies (cross-citation) for comparison and justification, what new results author obtained in this study or some new applications of previous work, verification of previous work, an improvement on previous work?

The impurities in phosphogypsum are influenced by the component composition of the raw materials used in the technological processes, it would be appropriate to consider this relationship.

In the conclusions (4) it would be appropriate to identify further directions of research and practical application for leaching of useful components of phosphogypsum.

Author Response

Dear Reviewer:

I am very grateful for your comments on the manuscript. According to your advice, we amended the relevant part of the manuscript, some of your questions were answered below.

-------------------------------------------------------------------------------------------------------------------------

Reply to the comments on molecules-1528525

Reviewer 1:

  1. The reviewer’s comments:

At the end of the Introduction section, the purpose and objectives of the study should be clearly identified. Example: “The research aims to …. To achieve this aim, the following tasks were set: ...”

The author’s answer:

Thank you for your valuable comments. We now correct the last sentence of the introduction as follows:

The research aims to reduce the cost of PG treatment and reduce environmental pollution risk, To achieve this aim, the following tasks were set: prensents a semi-dry method to solidify and stabilize PG.

  1. The reviewer’s comments:

In the used methods of testing (item 2.3) when mentioning the equipment on which the analysis of samples was carried out, it is necessary to write in brackets the manufacturers of this equipment and the country of the manufacturer. 

The author’s answer:

Thank you for your suggestion. We have modified it according to your comments. The details are shown in red in the revised version.

  1. The reviewer’s comments:

123, 177, 282, 288, 290, 292, 347, 351: The standards with which the author of the manuscript compared the results obtained are Chinese standards, is it possible to expand and specify also international standards for comparison?

The author’s answer:

Thank you for your suggestion, Due to the differences in composition and properties of phosphogypsum in different regions, Chinese standards are used for comparison in this study. Generally, Chinese standards are more stringent than international standards, and the raw material is Chinese phosphogypsum.

  1. The reviewer’s comments:

In the Results and discussion section (3) it is necessary when discussing the results obtained to make reference to previous studies (cross-citation) for comparison and justification, what new results author obtained in this study or some new applications of previous work, verification of previous work, an improvement on previous work?

The author’s answer:

Thank you for your valuable comments. In this study, the semi dry method is used for the first time to stabilize the solidification of toxic and harmful substances in phosphogypsum. There are few relevant studies, and some modifications have been made in the revised draft. See the revised draft for details.

  1. The reviewer’s comments:

The impurities in phosphogypsum are influenced by the component composition of the raw materials used in the technological processes, it would be appropriate to consider this relationship.

The author’s answer:

   Thank you for pointing out that the impurities in phosphogypsum are affected by the original phosphate rock and process, but the content of impurities is always small, and the content of calcium sulfate dihydrate is more than 80%. In the next research, we have considered the influence of impurities on raw materials and process.

  1. The reviewer’s comments:

In the conclusions (4) it would be appropriate to identify further directions of research and practical application for leaching of useful components of phosphogypsum.

The author’s answer:

Thank you for your valuable comments. The main purpose of this study is the solidification and stabilization of harmful components in phosphogypsum. As for the leaching of useful components in phosphogypsum is the treatment direction of high value-added phosphogypsum, our team will study it in the next work.

-------------------------------------------------------------------------------------------------------------------------

Many grammatical or typographical errors have been revised. All the lines and pages indicated above are in the revised manuscript.

Thank you and all the reviewers for the kind advice.

Sincerely yours,

Dr. Prof. Guangfei Qu.

Kunming University of Science and Technology.

Reviewer 2 Report

An important subject of research is the problem of the stability of phosphogypsum. They are considered to be highly toxic materials, the decomposition of which leads to the release of many toxic elements into the environment. Moreover, it is a problematic material in utilization due to its radioactivity and increased content of highly toxic elements. Such problems make the disposal of this material difficult. Many researchers have proposed different ways of doing this. However, some disposal suggestions were difficult to implement due to the problems described. The article presents an interesting method of stabilizing phosphogypsum and the idea of using it as filling in the workings of abandoned mines. Especially considering that phosphogypsum has a high sorption capacity and can quite often inhibit the leaching of toxic metals by chemisorption of some elements from the weathering zones, which can be formed in mines abandoned in mines. However, this should be the subject of research, as the development of the weathering zone in abandoned mines may lead to the re-destabilization of phosphogypsum.

The manuscript has the following problems:

  1. The language in the whole manuscript needs to be improved.
  2. Not all the shortcuts of names in the abstract are explained, which makes it difficult to read and understand the abstract. 
  3. In section 2.1 "Raw materials" the purity of chemical substances should be added. In addition, the manufacturer and model of the deionized water device should be also added. 
  4. Section 2.2 "Samples preparation" must be clarified e.g. shortcuts 1d, 3d, etc. I suspected that meant 1 day etc. but it is not clear. Moreover, it connects with a more serious mistake. In my opinion, the figures such as Fig.5, 6, 7, and 9, etc. suggested that experiments were conducted in 20 days (the X-axis on diagrams is not clear what means). The serious problem is also that the results should not be presented as points connected with lines (line charts). it also suggests that the figure presents changes in time, not a different experimental mixture. In my opinion, such charts should be presented as a bar chart.
  5. The methodology is not well described. XRF, SEM-EDS, FTIR methods is lacking important information. In the XRF section, there is no information on the operation of the lamp and the type of x-ray tube. The filter that was used for the analysis is not given and the time of collecting the spectrum is missing. There is no information on whether the samples were fused or pressed into pellets, if so, what the procedure looked like. There is no information as to whether the samples were homogenized and how it was done. In addition, whether the analyzes were performed with or without using standards. No field emission operating voltage in the SEM-EDS section. What was the working distance, aperture? Were the samples coated and, if so, with what thickness, and with what materials and equipment? The X-ray tube parameters for the analyzes that were performed are missing from the XRD. Which method was used for analysis Bragg-Brentano or DSH? Additionally, I noticed that on line 105 "... scan speed 80 per min ...". I suppose that's a mistake and the authors mean 8 degrees per minute. I propose to change the analysis step to 0.026 2Theta (1.56 degrees per minute) in the range from 4 to 80 2theta, which may allow detecting the less abundant phases in the samples. the FTIR methodology is missing information about the aperture, type of laser. In section 2.3.5 the information about the pH-meter manufacturer is missing.
  6. The methodology section is missing the information on how the grain size distribution, thermogravimetric (TG-DSC), and radioactivity were measured. 
  7. In the SEM-Images, I noticed that scale bare and annotations are illegible. Moreover, I noticed that the SEM-EDS analysis was conducted with 10 kV which is not good due to some heavier elements won't achieve the excited state with that energy. The EDS analysis should be conducted using at least 15 kV.  Moreover, in some cases at the EDS figures (Fig. 1, 8, and 12) the peaks name is missing or names form not important analytical lines are present. EDS figures do not look the same to avoid this problem I suggest preparing the figures in the excel program from saved EDS analysis in ASCII file. In addition, I wondered if the decomposition of overlapping peaks was conducted during calculating the Wt% form EDS measurements?
  8. The diffractograms are missing names over the peaks. 
  9. Table 3 should be ordered so that the same oxides are in the same columns.
  10. In my opinion, the control experiments are missing.
  11. The authors use the soluble salts terminology but the results are missing information on what elements are included in these salts.
  12. The mineral saturation index and the speciation in the experimental system need to be calculated based on AAS, ICP, and IC results. 
  13. It would be interesting to conduct such experiments in time (24h with intervals of sampling 5 minutes, 30 minutes, 1h, 2h, 5h, 10h, 15h, and 24h). 
  14. Units should be standardized throughout the article (e.g. in lines 337, 338 are used with different formatting of units).
  15. In the manuscript, the text should be used Fig. instead of Figure.
  16. The paper is lacking the proper discussion of the results and comparison them to literature data. I think that results should be a separate section in the article due to avoid confusion about which part is discussion and which is a results.

I hope that after correcting and adding a few analyses, calculations, and suggested changes, the revised manuscript will be suitable for publication.

Author Response

Dear Reviewer:

I am very grateful for your comments on the manuscript. According to your advice, we amended the relevant part of the manuscript, some of your questions were answered below.

-------------------------------------------------------------------------------------------------------------------------

Reply to the comments on molecules-1528525

Reviewer 2:

  1.  The reviewer’s comments:

The language in the whole manuscript needs to be improved.

The author’s answer:

We have revised the WHOLE manuscript carefully and tried to avoid any grammar or syntax error. In addition, we have asked several colleagues who are skilled authors of English language papers to check the English. We believe that the language is now acceptable for the review process.

  1.  The reviewer’s comments:

 Not all the shortcuts of names in the abstract are explained, which makes it difficult to read and understand the abstract. 

The author’s answer:

Thank you for your valuable comments. We have revised the summary and explained all abbreviations when they first appeared.

  1. The reviewer’s comments:

In section 2.1 "Raw materials" the purity of chemical substances should be added. In addition, the manufacturer and model of the deionized water device should be also added. 

The author’s answer:

Thank you for your valuable advice. Phosphogypsum and calcium carbide slag are mainly industrial by-products. The composition of calcium sulfate dihydrate in phosphogypsum is more than 80%, and the composition of calcium hydroxide in calcium carbide slag is about 80%. Only polyaluminium chloride and calcium chloride are chemically pure substances, both of which are analytical pure. The pure water machine (ULPHW-IV) for deionized water was purchased from Sichuan YOUPU ultra pure Technology Co., Ltd

  1.  The reviewer’s comments:

Section 2.2 "Samples preparation" must be clarified e.g. shortcuts 1d, 3d, etc. I suspected that meant 1 day etc. but it is not clear. Moreover, it connects with a more serious mistake. In my opinion, the figures such as Fig.5, 6, 7, and 9, etc. suggested that experiments were conducted in 20 days (the X-axis on diagrams is not clear what means). The serious problem is also that the results should not be presented as points connected with lines (line charts). it also suggests that the figure presents changes in time, not a different experimental mixture. In my opinion, such charts should be presented as a bar chart.

The author’s answer:

Thank you for your valuable comments. As your comments show, d stands for days, and we have revised it.

 The abscissa of figures 5, 6, 7 and 9 mainly represents the formula number. Figures 5, 6 and 7 correspond to the formula number in Table 1. and Figures 9 ~ 13 correspond to the formula number in Table 2. As the raw material of the main body is phosphogypsum, the concentration changes of pH, heavy metals and P, F, Cl, ammonia nitrate and water soluble salt under the conditions of adding different substances and different contents, so we think the lines (line charts) is more appropriate.

  1.  The reviewer’s comments:

The methodology is not well described. XRF, SEM-EDS, FTIR methods is lacking important information. In the XRF section, there is no information on the operation of the lamp and the type of x-ray tube. The filter that was used for the analysis is not given and the time of collecting the spectrum is missing. There is no information on whether the samples were fused or pressed into pellets, if so, what the procedure looked like. There is no information as to whether the samples were homogenized and how it was done. In addition, whether the analyzes were performed with or without using standards. No field emission operating voltage in the SEM-EDS section. What was the working distance, aperture? Were the samples coated and, if so, with what thickness, and with what materials and equipment? The X-ray tube parameters for the analyzes that were performed are missing from the XRD. Which method was used for analysis Bragg-Brentano or DSH? Additionally, I noticed that on line 105 "... scan speed 80 per min ...". I suppose that's a mistake and the authors mean 8 degrees per minute. I propose to change the analysis step to 0.026 2Theta (1.56 degrees per minute) in the range from 4 to 80 2theta, which may allow detecting the less abundant phases in the samples. the FTIR methodology is missing information about the aperture, type of laser. In section 2.3.5 the information about the pH-meter manufacturer is missing.

The author’s answer:

   Thank you for your valuable comments. We have given the detailed information of the instrument in the revised version, which is displayed in red font. Although you have put forward many parameters in the detailed instrument test process, this is not the focus of this study. XRD characterization is only to test the oxidation composition of raw materials. XRD is mainly to obtain the mineral phase information of raw materials, SEM-EDS is mainly to obtain the microstructure, FTIR is mainly to obtain the bonding and functional groups, and TG is mainly to conduct thermal stability analysis.

  1.  The reviewer’s comments:

The methodology section is missing the information on how the grain size distribution, thermogravimetric (TG-DSC), and radioactivity were measured. 

The author’s answer:

Thank you for pointing out, we have 2.3.5 particle size analysis and 2.3.6 thermogravimetric analysis in the test method (2.3) section.

  1.  The reviewer’s comments:

In the SEM-Images, I noticed that scale bare and annotations are illegible. Moreover, I noticed that the SEM-EDS analysis was conducted with 10 kV which is not good due to some heavier elements won't achieve the excited state with that energy. The EDS analysis should be conducted using at least 15 kV.  Moreover, in some cases at the EDS figures (Fig. 1, 8, and 12) the peaks name is missing or names form not important analytical lines are present. EDS figures do not look the same to avoid this problem I suggest preparing the figures in the excel program from saved EDS analysis in ASCII file. In addition, I wondered if the decomposition of overlapping peaks was conducted during calculating the Wt% form EDS measurements? 

The author’s answer:

Thank you for your valuable comments. We have made detailed modifications and attached the element content characterized by EDS in the revised draft.

  1.  The reviewer’s comments:

The diffractograms are missing names over the peaks.

The author’s answer:

Thank you for pointing out that we have made a revision.

  1.  The reviewer’s comments:

Table 3 should be ordered so that the same oxides are in the same columns.

The author’s answer:

This study is sorted according to the content, and we don't think there is much problem.

  1.  The reviewer’s comments:

In my opinion, the control experiments are missing.

The author’s answer:

This paper studies the solidification and stabilization experiment of phosphogypsum. The control experiment is the reaction system of phosphogypsum and calcium carbide slag.

  1.  The reviewer’s comments:

The authors use the soluble salts terminology but the results are missing information on what elements are included in these salts.

The author’s answer:

This paper studies the total soluble salt content, and the types of salt will be further considered in the future.

  1.  The reviewer’s comments:

The mineral saturation index and the speciation in the experimental system need to be calculated based on AAS, ICP, and IC results. 

The author’s answer:

Thank you for your valuable comments. This paper studies the solidification and stabilization of harmful components in phosphogypsum, and does not involve mineral saturation.

  1.  The reviewer’s comments:

It would be interesting to conduct such experiments in time (24h with intervals of sampling 5 minutes, 30 minutes, 1h, 2h, 5h, 10h, 15h, and 24h). 

The author’s answer:

Thank you for your valuable comments. This paper studies the semi dry reaction, which uses the self-contained water in phosphogypsum, which is not conducive to the reaction after a period of time.

  1.  The reviewer’s comments:

Units should be standardized throughout the article (e.g. in lines 337, 338 are used with different formatting of units).

The author’s answer:

Thank you for your reminder. We have unified all units of the article.

  1.  The reviewer’s comments:

In the manuscript, the text should be used Fig. instead of Figure.

The author’s answer:

Thank you for your reminder. I have replaced all the figures in this article with Fig.

  1.  The reviewer’s comments:

The paper is lacking the proper discussion of the results and comparison them to literature data. I think that results should be a separate section in the article due to avoid confusion about which part is discussion and which is a results.

The author’s answer:

Thank you for your valuable comments. We have quoted other people's studies for comparison in the results and discussion section, but we have not separated the results from the discussion. This part of this study needs to be discussed in combination with the results so that readers can better understand it.

-------------------------------------------------------------------------------------------------------------------------

Many grammatical or typographical errors have been revised. All the lines and pages indicated above are in the revised manuscript.

Thank you and all the reviewers for the kind advice.

Sincerely yours,

Dr. Prof. Guangfei Qu.

Kunming University of Science and Technology.

Round 2

Reviewer 2 Report

Dear authors, 

I think that the authors significantly improved the manuscript. They answered all my questions and dispelled all my doubts. However, I noticed one problem with citations in manuscript you use commas to separate cited works, which are consecutive (e.g. in line 45). In guides for authors is mentioned that in such cases should be cited as [9-11] not [9], [10], [11].